# Preoperative Prophylactic Balloon-Assisted Occlusion of the Internal Iliac Arteries in the Management of Placenta Increta/Percreta

**DOI:** 10.3390/medicina56080368

**Published:** 2020-07-23

**Authors:** Soo Buem Cho, Seok Jin Hong, Sangmin Lee, Jung Ho Won, Ho Cheol Choi, Ji Young Ha, Jin Il Moon, Ji Kwon Park, Ji Eun Park, Sung Eun Park

**Affiliations:** 1Department of Radiology, Ewha Womans University, College of Medicine, Seoul 07804, Korea; kingnose80@gmail.com; 2Department of Radiology, Gyeongsang National University Hospital, Jinju 52727, Korea; iamgoodbaba@hanmail.net (S.J.H.); lsmd10@naver.com (S.L.); circlehoya@naver.com (J.H.W.); jaro2@hanmail.net (H.C.C.); 3Department of Radiology, Gyeongsang National University Changwon Hospital, Changwon 51472, Korea; wonpiece@gmail.com (J.Y.H.); drlotus@naver.com (J.I.M.); 4Department of Obstetrics and Gynecology, Gyeongsang National University Changwon Hospital, Changwon 51472, Korea; obgypjk@gnu.ac.kr (J.K.P.); jl1104@daum.net (J.E.P.)

**Keywords:** prophylactic balloon-assisted occlusion, Apgar score, placenta increta, placenta accreta, placenta percreta, cesarean

## Abstract

*Background and Objectives*: Preoperative prophylactic balloon-assisted occlusion (PBAO) of the internal iliac arteries minimizes blood loss and facilitates surgery performance, through reductions in the rate of uterine perfusion, which allow for better control in hysterectomy performance, with decreased rates of bleeding and surgical complications. We aimed to investigate the maternal and fetal outcomes associated with PBAO use in women with placenta increta or percreta. *Material and Methods*: The records of 42 consecutive patients with a diagnosis of placenta increta or percreta were retrospectively reviewed. Of 42 patients, 17 patients (40.5%) with placenta increta or percreta underwent cesarean delivery after prophylactic balloon catheter placement in the bilateral internal iliac artery (balloon group). The blood loss volume, transfusion volume, postoperative hemoglobin changes, rates of hysterectomy and hospitalization, and infant Apgar score in this group were compared to those of 25 similar women who underwent cesarean delivery without balloon placement (surgical group). *Results*: The mean intraoperative blood loss volume in the balloon group (2319 ± 1191 mL, range 1000–4500 mL) was significantly lower than that in the surgical group (4435 ± 1376 mL, range 1500–10,500 mL) (*p* = 0.037). The mean blood unit volume transfused in the balloon group (2060 ± 1154 mL, range 1200–8000 mL) was significantly lower than that in the surgical group (3840 ± 1464 mL, range 1800–15,200 mL) (*p* = 0.043). There was no significant difference in the postoperative hemoglobin change, hysterectomy rates, length of hospitalization, or infant Apgar score between the groups. *Conclusion*: PBAO of the internal iliac artery prior to cesarean delivery in patients with placenta increta or percreta is a safe and minimally invasive technique that reduces the rate of intraoperative blood loss and transfusion requirements.

## 1. Introduction

Morbidly adherent placenta (MAP) occurs when all or part of the chorionic villi attaches abnormally to the myometrium. Three grades of abnormal placental attachment have been defined according to the depth of attachment and invasion into the myometrium: accreta (chorionic villi attach to the myometrium), increta (chorionic villi invade the myometrium), and percreta (chorionic villi penetrate the uterine serosa) [1]. Patients with MAP are at increased risk of peripartum complications, including massive bleeding, hysterectomy, and even death [2]. Intraoperative blood loss may require significant blood transfusion, accompanied by disseminated intravascular complications, fluid overload, and acute respiratory distress syndrome. Although the impact of MAP on pregnancy outcomes has been well-described, to the best of our knowledge, no prospective randomized trial and few studies overall have examined the management of pregnancies complicated by this disorder.

The appropriate management of MAP is an issue of critical concern to obstetricians. The presence of a previous history of cesarean section is associated with a high risk of MAP. The increasing MAP prevalence and severity of the related adverse outcomes have attracted high levels of attention, especially in cases with placenta increta and percreta [3]. In this study, we sought to focus on placenta increta and percreta. Despite the guideline published by the American College of Obstetricians and Gynecologists [4], which recommends the performance of planned preterm cesarean hysterectomy while leaving the placenta in situ for MAP management, the achievement of uterus preserving management is being continuously reported on in the application of different methods to reduce bleeding rates, such as prophylactic balloon-assisted occlusion (PBAO), uterine artery embolization (UAE), uterine artery ligation, uterine compression suture, and uterine tamponade [5]. These conservative managements may be applied for some women who want to be able to have more children.

Preoperative PBAO of the internal iliac arteries is performed with the aim of minimizing the blood loss volume and facilitating surgery. The theory is that the reduction of the rate of uterine perfusion allows for better control in hysterectomy performance, with decreased rates of bleeding and surgical complications. Patrick et al. [6] reported that this approach is safe for both mother and baby. Conversely, some studies reported equivocal results and highlighted the potentially serious complications associated with PBAO [7,8]. However, the existing literature on this approach is limited because of the mixed occurrence of placenta accreta, increta, and percreta in patient populations.

This study aimed to compare the maternal and fetal outcomes of those who underwent cesarean delivery for placenta increta or percreta, with and without the use of PBAO.

## 2. Materials and Methods

We retrospectively reviewed the medical records and radiologic findings of 42 consecutive patients with a diagnosis of placenta increta or percreta between April 2010 and May 2019. The Institutional Review Board of Gyeongsang National University Changwon Hospital approved this study, and the need for informed consent was waived because of retrospective nature of this study. Elective cesarean section was performed in all cases, and preoperative diagnoses were confirmed using transabdominal and transvaginal sonography and Doppler sonography; magnetic resonance imaging was used as a problem-solving tool only in cases with an unclear diagnosis. The imaging criteria suggestive of placenta increta/percreta included (1) multiple lacunar flow pattern, (2) evidence of invasion of the inner half of the myometrium, (3) the interruption of uterine wall and bladder interface, (4) interface hypervascularity with abnormal blood vessels linking the placenta to the bladder, and (5) markedly dilated peripheral subplacental vascular channels. The final diagnosis of any placenta increta or percreta was confirmed based on the degree of placental invasion, according to the histopathological examination of placental bed biopsies or hysterectomy spcimens.

The patients were subdivided into categories based on the use of PBAO in the bilateral internal iliac arteries (balloon group), before the cesarean delivery in the operating room. All the PBAO procedures were performed in the interventional suite by three interventional radiologists with 4–12 years of experience. The proximity between the operating room and interventional suite allowed for a smooth transition between PBAO and subsequent cesarean delivery. The control group included those who had undergone elective cesarean delivery alone (surgical group) without PBAO. The surgical group comprised patients with similar diagnoses from the same institutions during the same study time. Unexpected emergency cesarean delivery was excluded to maintain homogeneity between the two groups.

On the day of delivery, patients were taken to the interventional suite for PBAO, bilateral common femoral arterial puncture, and access of the 6-Fr vascular sheaths under local anesthesia. Both the internal iliac arteries were cannulated using a contralateral approach, with a 5-Fr reverse curve catheter (Yashiro; Terumo, Tokyo, Japan or UAC; Merit Medical, Jordan, UT, USA). This was exchanged for a 6- or 7-mm-diameter, 40-mm-long, standard percutaneous transluminal angioplasty balloon catheter (Mustang; Boston Scientific, Galway, Ireland), which was positioned with its tip in the proximal portion of the contralateral internal iliac artery, just after the common iliac artery bifurcation (Figure 1). Arterial diameter measurement was performed at the time of balloon insertion to ensure correct balloon sizing. We found that 1 mL (6-mm × 40-mm) and 1.5 mL (7-mm × 40-mm) of contrast media were generally sufficient for artery occlusion. The exact same volume was subsequently used to inflate the balloons, hence eliminating the need for additional fluoroscopic exposure during cesarean delivery. The catheters were securely taped to the skin, and the sheaths were sutured in place, with a transparent dressing placed over them to further ensure stability.

In order to reduce the radiation dose to the mother and fetus, all possible measures were taken. The fluoroscopy time was kept to a minimum, and the radiation dose was maintained in accordance with the ALARA (As Low As Reasonably Achievable) principle. Last-image hold was used wherever possible instead of digital subtraction angiography, and spot X-ray images avoided. The X-ray tube was kept as far away as possible from the patient, while the detector was kept as close as possible. No magnification was used, while tight collimation was employed wherever possible. Data on all radiation-related factors, including fluoroscopy time, dose area product (DAP), kVp, and air kerma were recorded at the time of PBAO.

The patient was then transferred to the operating room for cesarean delivery. After delivery, the balloons were inflated at the time of cord clamping to minimize the risk of fetal ischemia. In patients undergoing hysterectomy or delivery of the placenta, the balloons were deflated just before the skin closure after ensuring that hemostasis within the pelvic cavity was secured. Among patients in whom the placenta was retained or in those with continuous bleeding, the balloons were left inflated until UAE performance.

The degree of intraoperative blood loss, which was the estimated amount of blood loss as calculated with reference to the contents of the suction apparatus and to weight of the surgical pads, and volume of packed red blood cells received were compared between the two groups. The immediate preoperative (the morning of cesarean delivery) and postoperative (immediately after cesarean delivery) hemoglobin levels, hysterectomy rates, lengths of hospitalization, and infant Apgar scores were also compared between the groups. Study data were tested for normal distribution using a Kolmogorov–Smirnov test. Normally distributed variables were compared using independent *t*-tests, and presented as mean ± standard deviation. Categorical variables were compared using the chi-square test or Fisher’s exact test for small cell values. All statistical analyses were performed using SPSS (IBM SPSS Statistics for Windows, Version 24.0. Armonk, NY: IBM Corp), and statistical significance was considered at a probability value lower than 0.05.

## 3. Results

A total of 17 patients in the balloon group were compared with 25 patients in the surgical group. Data on the maternal demographic characteristics and clinical outcomes in both groups are summarized in Table 1 and Table 2. The two groups of patients showed similar values in terms of their mean maternal age, gravity and parity status, history of cesarean section, and pathology of MAP (Table 1). None of the parameters listed in Table 1 showed significant differences between the groups. Table 2 presents the comparison of the blood loss parameters, rates of hysterectomy, lengths of hospitalization, and infant Apgar scores. These included the mean estimated intraoperative blood loss and mean volume of packed red blood cells transfused during surgery. The levels of both the latter parameters were significantly lower in the balloon group than the surgical group (*p* < 0.05). There was no significant difference in the mean change in the hemoglobin level, rates of hysterectomy, lengths of hospitalization, and infant Apgar scores between the groups. All patients with placenta percreta in both groups underwent hysterectomy. Although not statistically significant, the balloon group showed higher rates of uterine preservation (12/14, 85.7%) than the surgical group (14/21, 66.7%) when only patients with placenta increta were compared. Twenty-four of the 42 patients (57.1%) underwent additional UAE after delivery. The number of patients who underwent UAE after delivery did not significantly differ between the balloon group (9/17, 52.9%) and surgical group (15/25, 60%) (*p* = 0.206).

The overall mean DAP of the patients who received PBAO was 262.8 µGym^2^ (range, 180.8–361.2 µGym^2^). The mean cumulative fluoroscopy time and mean cumulative air kerma in the balloon group were 159.2 s (range, 121–218 s) and 45.78 mGy (range, 32.18–63.79 mGy), respectively. These values did not include the degree of radiation dose exposure during additional UAE.

No still births or maternal death were observed in either group. No vascular complication occurred related to PBAO in the balloon group. A total of 44 live infants were delivered in both groups, including two sets of twins. The mean infant Apgar scores at 1 and 5 min were 7.0 and 8.3, respectively. None of the infants developed respiratory distress syndrome, intracranial hemorrhage, evidence of hypoxic ischemic injury, or necrotizing enterocolitis, and none of them required follow-up neurological imaging during hospitalization.

## 4. Discussion

This study showed that PBAO is an effective adjunct to cesarean delivery for the minimization of blood loss in patients with placenta increta or percreta. Patients with MAP are at a high risk of life-threatening hemorrhage, with obstetric hemorrhage known to be a major cause of maternal mortality. Although the impact of MAP on pregnancy outcomes has been well-described, to the best of our knowledge, no prospective randomized trial and few studies overall have examined the management of pregnancies complicated by this disorder. Recent studies showed that PBAO reduced the rate of blood loss and transfusion requirement in patients with MAP [6,9,10]. Nicholson et al. reported that patients with PBAO had a decreased rate of hysterectomy compared to those without it. However, other studies showed conflicting results concerning the efficacy of PBAO; Shrivastava et al. reported that PBAO did not benefit women with MAP undergoing cesarean hysterectomy, while Sentilhes et al. demonstrated that conservative management (UAE, uterine artery ligation, uterine compression suture, and uterine tamponade) without PBAO helped in the avoidance of hysterectomy, and involved a low rate of severe maternal morbidity. In the two aforementioned studies, pathologically confirmed placenta accreta accounted for 68% and 89.2% of the total MAP cases. Regarding the PBAO technique, 37 reports presenting one or more placenta increta or percreta cases have been published. Heterogeneity characterized these case series, and they included not only patients with placenta increta and percreta, but also those with placenta accreta and previa. Furthermore, most case series include only one or two placenta percreta series—the condition associated with the highest risk of maternal and fetal morbidity and mortality. In clinical practice, the exact MAP classification criteria still vary from study to study, especially in terms of placenta accrete [3]. Although histological examination is recommended, it may not be accurate when the uterus is conserved. Indeed, simple placenta accreta is most often resolved by manual delivery for incomplete separation. Placenta increta or percreta are less commonly observed than placenta accrete, but are associated with greater morbidity, as well as maternal and fetal mortality. Thus, we focused on placenta increta and percreta, in the current study.

Hudon et al. [5] reported in patients with placenta percreta an average intraoperative blood loss volume of 3000–5000 mL, and that 90% of patients undergoing cesarean-hysterectomy will require transfusion. Chantraine et al. [11] showed in a recent case series of placenta increta and percreta an average red blood cell volume of 2800 mL. In the present study, the mean estimated intraoperative blood loss and mean volume of packed red blood cells in the balloon group were 2319 mL and 2060 mL, respectively. The levels of both parameters were significantly lower in the balloon group than the surgical group. The investigators, who failed to identify benefits associated with PBAO, suggested that the mean blood loss did not decrease as a result of the developed collateral blood supply to the uterus. They proposed that the performance of prophylactic embolization before balloon deflation is essential for better postpartum bleeding control [12]. In the present study, we performed additional UAE after delivery only if the gynecologist determined that postpartum bleeding persisted and was necessary. There was no significant difference in the number of patients who underwent additional UAE between the balloon group and surgical group. The optimal placement site of the balloon catheter can be a critical factor affecting outcomes and side effect incidence [13]. In most studies, the anterior division of the internal iliac artery was used as the side of ballooning for vessel occlusion, but positions as proximal as the abdominal aorta and as distal as those within the anterior division of the internal iliac arteries have been employed [1,2,6,7,8,9,10,12,13,14,15,16,17,18]. We used the proximal part of the internal iliac artery just after the common iliac artery bifurcation; this approach reduces the intervention time, collateral blood flow rate, and fetal radiation exposure degree more than occlusion of the anterior division of the internal iliac artery [17].

Surgical internal iliac artery ligation is used to control the degree of intractable postpartum hemorrhage. Previous data have shown that the success rates of surgical ligation of the uterine or internal iliac arteries varied widely from 40% to 100% [19]. Although the number of cases is limited, the success rate of PBAO is 100% in most studies [1,6,13,14,17]. Obstetric hemorrhage related to MAP tends to occur during delivery or the postpartum period [20]. Therefore, for patients who are likely to have severe postpartum hemorrhage, the performance of PBAO prior to cesarean delivery can minimize the blood flow volume through immediate balloon inflation. Since balloon catheters and sheaths are inserted, it is easy to perform additional uterine artery embolization immediately in the case of uncontrolled bleeding after cesarean delivery. In contrast, bleeding control may take longer with surgical ligation use; thus, the prophylactic use of bilateral internal iliac artery ligation has been found to have limited advantages in the reduction of inter-operative bleeding rates [21].

For many years, hysterectomy was the treatment of choice for patients with these conditions because of the unacceptably high maternal mortality values. The use of conservative treatment without hysterectomy is increasingly being advocated when the degree of blood loss is not excessive, defects are focal, and future fertility is desired [22]. Low mortality values have been reported in carefully selected patients [22,23], while another study has shown the opposite [24]. Nicholson et al. [6] reported that patients with PBAO had decreased rates of hysterectomy compared to the historic controls, which is inconsistent with our findings. However, in that study, only two and one patients had a diagnosis of placenta increta and percreta, respectively, of 22 patients with MAP who underwent PBAO. According to a retrospective study of 17 patients with placenta percreta who underwent PBAO, the rate of hysterectomy use was 47.1% (8/17) [25]. In a study by Angileri et al. [26], uterine preservation was achieved in all 20 patients with placenta percreta. Various factors such as differences in the skills, training environment, and experience of gynecologists contribute to the wide hysterectomy use range. A prospective survey involving a large number of placenta increta and percreta cases was limited by the occurrence rarity of MAP. Therefore, further prospective, randomized studies in multicenter settings will be needed to determine if PBAO contributes to lowering the rate of hysterectomy.

We paid particular attention to the minimization of fetal radiation exposure during the procedure, using appropriate shielding and intermittent low-dose fluoroscopy. There is no evidence stating that radiation doses lower than 200 mGy are associated with an increased incidence of fetal congenital malformation in humans [18]. In our study, the mean DAP of patients who received PBAO was 262.8 µGym^2^, which was much lower than the radiation dose of 200 mGy. A major concern is the potential oncogenic effect of radiation to the fetus. In studies showing a mean DAP similar to that used in our study, the average fetal radiation dose was between 3.2 and 6.3 mGy [17,18,27]. It has been estimated that there is a 99.5% likelihood of a child never developing childhood cancer following exposure to fetal radiation doses up to 100 mGy.

This study has several limitations. Unavoidable selection bias was present, because of the study’s single-center retrospective design, as well as the relatively small sample size for the comparison of statistically significant differences between the groups. The lack of randomization is an issue, as the more complex cases may have been selected for PBAO while the less severe cases may have likely undergone standard cesarean delivery.

In summary, placenta increta and percreta are potentially life-threatening hemorrhage conditions. The use of PBAO during cesarean delivery offers several advantages. Bilateral PBAO aids in the achievement of hemodynamic stability and ensuring optimal exposure of the pelvic organs during surgery. The potential need for transfusion is, therefore, reduced. Thus, PBAO may not only limit the rate of blood loss, but may also minimize the risk of transfusion reactions and blood-borne illnesses such as hepatitis B and C and human immunodeficiency virus infection. The technical aspects of the procedure are straightforward, with minimal procedure-related complications. In addition, it is performed quickly with minimal fetal radiation exposure, which in this study was 121–218 s of cumulative fluoroscopy time.

## 5. Conclusions

In conclusion, the performance of PBAO in patients with placenta increta or percreta undergoing delivery is effective in decreasing the amount of intraoperative blood loss and rate of transfusion requirements compared to the control group. In our experience, it is a safe and minimally invasive technique for both mother and baby.

## Figures and Tables

**Figure 1 medicina-56-00368-f001:**
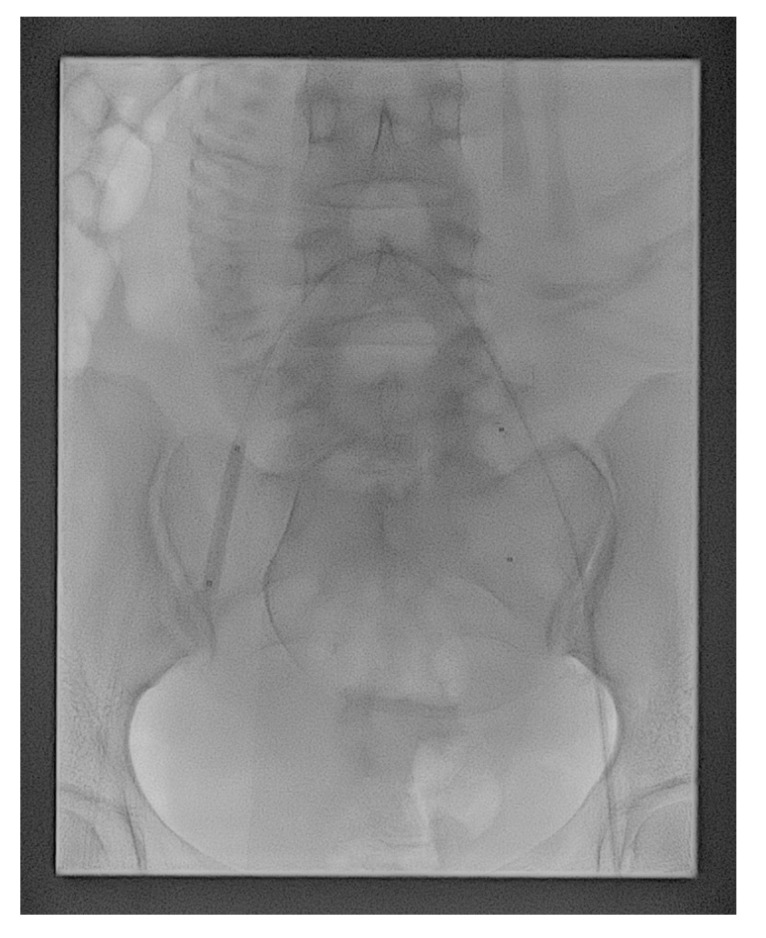
Fluoroscopy showing the balloon catheters positioned in the main lumen of the internal iliac arteries bilaterally. The balloon catheter of the right internal iliac artery was inflated as a test.

**Table 1 medicina-56-00368-t001:** Demographic characteristics and incidence of different forms of morbidly adherent placenta (MAP) in the two study groups.

Parameter	Balloon Group (N = 17)	Surgical Group (N = 25)	*p* Value
Mean maternal age (y) (range)	34.5 ± 2.9	34.8 ± 3.3	NS
Mean gravity	2.2 ± 1.2	1.9 ± 0.9	NS
Mean parity	1.4 ± 0.7	1.3 ± 0.6	NS
Previous cesarean section	0.47 ± 0.8	0.29 ± 0.6	NS
Placenta pathology			
Increta	14 (82.4%)	21 (84%)	NS
Percreta	3 (17.6%)	4 (16%)	NS

Note, Values are shown as mean ± standard deviation or *n* (%), NS: not significant.

**Table 2 medicina-56-00368-t002:** Clinical outcomes in the two study groups.

Parameter	Balloon Group (N = 17)	Surgical Group (N =25)	*p* Value
Estimated blood loss (mL) (range)	2319 (1000−4500)	4435 (1500−10,500)	0.037 *
Transfused packed RBCs (mL) (range)	2060 (1200−8000)	3840 (1800−15,200)	0.043 *
Mean hemoglobin change (g/dL) (range)	2.1 (0.7−3.4)	2.8 (0.9−5.1)	0.251
Total hysterectomies	5 (29.4%)	11 (44%)	0.339
Hysterectomies/Increta	2/14 (14.3%)	7/21 (33.3%)	0.207
Hysterectomies/Percreta	3/3 (100%)	4/4 (100%)	NS
Hospital stay (days) (range) Wound infection Bladder injuries Maternal mortality	10.2 (6−11) 1 (5.9%) 0 0	10.8 (5−33) 3 (12%) 1 (4%) 0	0.767 0.591 NS NS
Apgar scores-1 min (range)	6.7 (5−8)	7.2 (5−9)	0.560
Apgar scores-5 min (range)	8.1 (7−9)	8.5 (7−10)	0.471

Note, Values are shown as mean ± standard deviation or *n* (%), RBCs: red blood cells, NS: not significant, * *p* < 0.05.

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
