# Peer review of "Preoperative Prophylactic Balloon-Assisted Occlusion of the Internal Iliac Arteries in the Management of Placenta Increta/Percreta"

_medicina, 2020, doi:10.3390/medicina56080368_

Round 1

Reviewer 1 Report

thanks to the authors, they have addressed all the issues raised by reviewers. Congratulations for your work!

Reviewer 2 Report

The authors of the manuscript correctly addressed the reviewers' comments.

This manuscript is a resubmission of an earlier submission. The following is a list of the peer review reports and author responses from that submission.

Round 1

Reviewer 1 Report

Placenta accreta/percreta is a leading cause of third trimester hemorrhage and postpartum maternal death. The current treatment for third trimester hemorrhage due to morbidly adherent placenta is cesarean hysterectomy, which may be complicated by large volume blood loss.

In literature are present divergent recommendations for the use of preoperative prophylactic ballon-assisted occlusion of the internal iliac arteries in the management of patients with abnormal placentation. The authors share their experience with this pre- operative modality. 

The paper is well written, with a clear text easy to read. tables are usefol to immediate understand the paper results. The only limit i would like to highligt is the limited number of examined cases. Unfortunately too little to be able to obtaim strong conclusive considerations.

Best regards.

Reviewer 2 Report

The authors of this retrospective study investigated the maternal and fetal outcomes associated with preoperative prophylactic balloon-assisted occlusion (PBAO) use in 42 women with placenta increta or percreta undergoing cesarean delivery. The results of the experimental group were compared to 25 similar women who underwent cesarean delivery without PBAO. Globally, PBAO prior to cesarean delivery in patients with placenta increta or percreta resulted a safe and minimally invasive technique. Notably, it was effective in decreasing the amount of intraoperative blood loss and rate of transfusion requirements compared to the control group. The topic is of interest; however, this manuscript has some issues:

- which was the study time?

- has been done a power calculation for deciding the sample size of each group? If yes, can it be reported in the material and methods.

- the inclusion of only elective cesarean section should be better under scored in the material and methods.

- it should be better described by which criteria the diagnosis of placenta accrete/increta/percreta was done at ultrasound and/or magnetic resonance. Was in all the cases confirmed at pathological analysis?

- the term blood product should be avoided and the results should be also also separately reported for transfusion of red cells/plasma.

- how to do you evaluate blood loss? When exactly do you evaluated postoperative hemoglobin?

Minor comments

Abstract ln 35 use the abbreviation PBAO